# TRANSLATOTRON 2:
# ROBUST DIRECT SPEECH-TO-SPEECH TRANSLATION

## ABSTRACT

We present *Translatotron 2*, a neural direct speech-to-speech translation model that can be trained end-to-end. Translatotron 2 consists of a speech encoder, a phoneme decoder, a mel-spectrogram synthesizer, and an attention module that connects all the previous three components. Experimental results suggest that Translatotron 2 outperforms the original Translatotron by a large margin in terms of translation quality and predicted speech naturalness, and drastically improves the robustness of the predicted speech by mitigating over-generation, such as babbling or long pause. We also propose a new method for retaining the source speaker's voice in the translated speech. The trained model is restricted to retain the source speaker's voice, but unlike the original Translatotron, it is not able to generate speech in a different speaker's voice, making the model more robust for production deployment, by mitigating potential misuse for creating spoofing audio artifacts. When the new method is used together with a simple concatenation-based data augmentation, the trained Translatotron 2 model is able to retain each speaker's voice for input with speaker turns.

## 1 INTRODUCTION

Speech-to-speech translation (S2ST) is highly beneficial for breaking down communication barriers between people not sharing a common language. Conventional S2ST systems are often composed of a cascade of three components: automatic speech recognition (ASR), text-to-text machine translation (MT), and text-to-speech (TTS) synthesis (Lavie et al., 1997; Wahlster, 2000; Nakamura et al., 2006). Very recently, direct speech-to-text translation (ST) is rapidly emerging, and has outperformed the cascade of ASR and MT (Weiss et al., 2017; Jia et al., 2019a; Di Gangi et al., 2019; McCarthy et al., 2020; Wang et al., 2021b; Ansari et al., 2020; Anastasopoulos et al., 2021), which makes the cascade of ST and TTS as S2ST feasible (Jia et al., 2019b). However, works on direct S2ST is very limited.

Compared to cascaded systems, direct S2ST has the potential benefits of: 1) retaining paralinguistic and non-linguistic information during the translation, such as speaker's voice (Jia et al., 2019b), emotion and prosody; 2) working on languages without written form (Tjandra et al., 2019; Zhang et al., 2021; Lee et al., 2021a); 3) reduced computational requirements and lower inference latency; 4) avoiding error compounding across sub-systems; 5) easier on handling contents that do not need to be translated, such as names and proper nouns (Jia et al., 2019b).

Translatotron (Jia et al., 2019b) is the first model that is able to directly translate speech in one language to speech in another language. It is also able to retain the source speaker's voice in the translated speech. However, its translation quality underperforms a cascade baseline by a large margin, and its synthesized translated speech suffers from robustness issues, such as babbling and long pause, which is an issue shared with the Tacotron 2 TTS model (Shen et al., 2018; He et al., 2019; Zheng et al., 2019; Battenberg et al., 2020; Shen et al., 2020), since they share the same attention-based approach for synthesizing speech.

In this work we present *Translatotron 2*. The main contributions include:

1. We propose *Translatotron 2*, a novel direct S2ST model that is able to be trained end-to-end;
2. We conduct experiments suggesting that Translatotron 2 significantly outperforms Translatotron, and is comparable to a cascade system, in terms of translation quality, speech naturalness, and speech robustness;

3. We propose a new method for voice retention in S2ST without relying on any explicit speaker embedding or ID. The trained model is only able to retain the source speaker's voice but not able to generate speech in a different speaker's voice, making it more robust for production deployment by mitigating potential misuse for creating spoofing audio artifacts;

4. We propose a simple concatenation data augmentation, *ConcatAug*, to enable Translatotron 2 to retain each speaker's voice when the input includes speaker turns;

5. We conduct experiment suggesting that Translatotron 2 is efficient on multilingual direct S2ST, in which it obtaines translation quality very close to an ST baseline.

Audio samples from Translatotron 2 are available in the supplementary material.

## 2  RELATED WORKS

**S2ST**   Until very recently, S2ST systems are typically composed of a cascade of ASR, MT, and TTS components (Lavie et al., 1997; Wahlster, 2000; Nakamura et al., 2006; ITU, 2016). Translatotron (Jia et al., 2019b) is the first direct S2ST model, which is a sequence-to-sequence model trained end-to-end in a multi-objective task. It has shown reasonable translation quality and speech naturalness, but still underperformed a baseline of ST + TTS cascade by a large margin. It also demonstrated the capacity of retaining speaker's voice during the translation, by leveraging a speaker encoder separately trained in a speaker verification task (Wan et al., 2018; Jia et al., 2018).

A few recent works proposed cascade S2ST systems using learned discrete representation as the intermediate representation instead of text or phoneme. Tjandra et al. (2019) introduced such an S2ST system that first translated the source speech into a discrete representation of the target speech which was predicted from a separately trained VQ-VAE (Oord et al., 2017), and then used the VQ-VAE decoder to predict the target speech spectrogram from the discrete representation. Zhang et al. (2021) additionally trained the VQ-VAE jointly with a supervised phoneme recognition objective in different languages. Lee et al. (2021a) used a separately trained vocoder to directly predict waveform from the discrete representation without relying on spectrogram; for the best performance, this vocoder included a duration predictor and an upsampler, akin to a generative TTS model. All these works require multiple components being trained in multiple steps, but are not able to be trained end-to-end.

Kano et al. (2021) introduced an end-to-end S2ST model with a cascade of three auto-regressive decoders, and used pre-trained MT and TTS models as teacher models to facilitate the training of the end-to-end model. It requires pre-trained ASR, MT, and TTS models, and the end-to-end model itself has to be trained in multiple steps.

Unfortunately, despite that these recent works generated translated speech in novel ways without relying on TTS subsystems, most of these works (except for Jia et al. (2019b)) focused only on the translation quality, but did not assess the perceptual quality (Wagner et al., 2019) of the translated speech (e.g. naturalness), which is critical to S2ST.

**TTS**   Translatotron uses a decoder similar to Tacotron 2 (Shen et al., 2018; Jia et al., 2018), which is an attention-based auto-regressive decoder. Due to the flexibility of the attention mechanism, they both suffer from robustness issues such as over-generation. Recent TTS models such as FastSpeech (Ren et al., 2019; 2021), Non-Attentive Tacotron (NAT) (Shen et al., 2020; Jia et al., 2021) and Parallel Tacotron (Elias et al., 2021b;a), demonstrate that replacing the attention module with a duration-based upsampler yields more robust synthesized speech, as quantitatively evaluated at a large scale in Shen et al. (2020). The synthesizer component in this work resembles these works.

**Voice conversion and anti-spoofing**   The performance of voice conversion (VC) has progressed rapidly in the recent years, and is reaching a quality that is hard for automatic speaker verification (ASV) systems to detect (Yi et al., 2020). ASVspoof 2019 (Todisco et al., 2019; Wang et al., 2020) found that it was challenging to detect spoof audios generated from Jia et al. (2018), which uses the same speaker encoder-based approach as in the original Translatotron. Such progress poses concerns on related techniques being misused for creating spoofing artifacts. We designed Translatotron 2 with the motivation of avoiding such potential misuse.

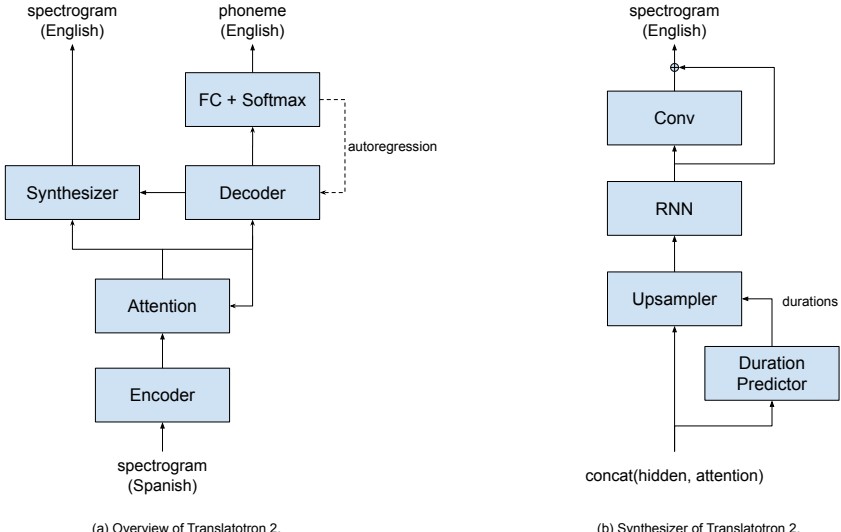

(a) Overview of Translatotron 2.

(b) Synthesizer of Translatotron 2.

Figure 1: A Translatotron 2 model that translates Spanish speech into English speech.

# 3 TRANSLATOTRON 2

The motivation of the architectural design of Translatotron 2 is to improve on three weaknesses existing in the original Translatotron: 1) The valuable auxiliary supervision on textual features is not fully utilized during training; 2) Long-sequence-to-long-sequence modelling with attention is difficult because of the flexibility of the attention mechanism; 3) Attention-based speech generation is known to suffer from robustness issues such as over-generation and under-generation.

The proposed *Translatotron 2* model solves these problems by using an architecture composed of a source speech encoder, a target phoneme decoder, and a target mel-spectrogram synthesizer. These three components are connected together by a single attention module. See Figure 1 for an illustration. The model is jointly trained with a speech-to-speech translation objective and a speech-to-phoneme translation objective. As the result, the auxiliary supervision is fully utilized, and the attention is used only for modeling shorter target phoneme sequence but is not directly involved in speech generation.

The following subsections describe the architecture of each components as used in our main experiments. As shown in Appendix D, using alternative architectures for these components do not significantly impact the performance of Translatotron 2.

## 3.1 ENCODER

The encoder of Translatotron 2 takes mel-spectrogram of the source speech as the input, and produces a hidden representation. We use Conformer (Gulati et al., 2020) as the architecture for the encoder. It first subsamples the input mel-spectrogram with a convolutional layer, and then processes it with a stack of Conformer blocks. Each Conformer block is composed of a feed-forward layer, a self-attention layer, a convolution layer, and a second feed-forward layer. SpecAugment (Park et al., 2019) is applied at the training time.

## 3.2 DECODER

The autoregressive decoder is assisted with an attention module. It takes the encoder output as the source values for the attention, and predicts phoneme sequences corresponding to the target translated speech. We use a stack of LSTM cells as the decoder, along with a multi-head attention (Vaswani et al., 2017). The combination of the encoder, the decoder, and the attention module is similar to a typical ST model, except that it predicts phonemes instead of subword tokens.

### 3.3 Synthesizer

The synthesizer takes the hidden output from the decoder, as well as the context output from the attention module as its input, and synthesizes the target mel-spectrogram. It is similar to the decoders in typical neural TTS models, such as in Shen et al. (2018; 2020); Ren et al. (2021). The predicted mel-spectrogram can be converted into waveform using either an estimation algorithm such as Griffin & Lim (1984) or a neural vocoder.

We experimented with a number of different non-attentive architectures for the synthesizer, including autoregressive ones and parallel ones, and found that autoregressive synthesizers worked best. We followed the architecture and the hyperparameters of the mel-spectrogram decoder from NAT (Shen et al., 2020). It first predicts a duration for each of its input elements using a bidirectional LSTM stack, then upsamples the input sequence with Gaussian weights based on the predicted duration, and finally uses an auto-regressive LSTM stack to predict the target mel-spectrogram. The predicted mel-spectrogram is further refined with a residual convolutional post-net. This synthesizer is trained jointly with a mel-spectrogram reconstruction loss and a duration prediction loss. Figure 1(b) illustrates the architecture of this synthesizer.

Unlike in Shen et al. (2020), we do not use per-phoneme duration labels for training the duration predictor, nor do we apply the FVAE-based alignment. Instead, only an $L^2$ loss on the total predicted duration of the entire sequence is used (i.e. the "naïve approach" of unsupervised duration modelling in Shen et al. (2020)). We anticipate that adopting these approaches could further improve the performance of Translatotron 2, which we leave as future work.

## 4 Voice retention

The original Translatotron (Jia et al., 2019b) demonstrated the capacity of being able to retain the source speaker's voice in the translated speech, by conditioning its decoder on a speaker embedding generated from a separately trained speaker encoder. In fact, it is capable of generating the translated speech in a different speaker's voice, as long as a clip of the target speaker's recording is used as the reference audio to the speaker encoder, or the embedding of the target speaker is directly available. While this is impressively powerful, it can potentially be misused for generating spoofing audio with arbitrary content, posing a concern for production deployment.

In this work, we use a different approach for retaining the source speaker's voice in the translated speech. The trained model is restricted to retaining the source speaker's voice, but not able to generate speech in a different speaker's voice.

### 4.1 Training targets in source speaker's voice

To retain speakers' voices across translation, we train S2ST models on parallel utterances with the same speaker's voice on both sides. Such a dataset with human recordings on both sides is extremely difficult to collect, because it requires a large number of fluent bilingual speakers. Instead, we use a TTS model that is capable of cross-lingual voice transferring to synthesize such training targets.

We modified the PnG NAT (Jia et al., 2021) TTS model by incorporating a separately trained speaker encoder (Wan et al., 2018) in the same way as Jia et al. (2018), and trained it on the LibriTTS corpus (Zen et al., 2019). The result TTS model is capable of zero-shot voice transferring, but synthesizes in a better quality and more robust than Jia et al. (2018). We used this model to synthesize target speech in the source speaker's voice in our experiments. Other TTS models capable of cross-lingual voice modelling, such as Zhang et al. (2019); Chen et al. (2019); Xin et al. (2021), could also be utilized.

### 4.2 Speaker turns

Theoretically, because the target-spectrogram synthesizer in both Translatotron 2 and Translatotron are directly conditioned on the source-spectrogram encoder output, the encoder output may be capable of preserving voice information locally in together with linguistic information, and the decoders and synthesizers may be capable of utilizing such local information for translating linguistic information while preserving local voice information. As a result, such direct S2ST models may be capable of

Table 1: Datasets for experiments with a single-speaker target.

|  | Conversational (Jia et al., 2019a) | Fisher (Post et al., 2013) | CoVoST 2 (Wang et al., 2021a) |
|---|---|---|---|
| Languages | es→en | es→en | es, fr, de, ca → en |
| Domain | Read, short-form | Telephone conversation | Read, short-form |
| Source sample rate | 16-48 kHz | 8 kHz | 48 kHz |
| Utterance pairs | 979k | 120k | 321k |
| Source hours | 1,400 | 127 | 476 |
| Target hours | 619 | 96 | 296 |
| Target synthesized by | Tacotron 2 + Griffin-Lim | Parallel WaveNet | PnG NAT + WaveRNN |

retaining each source speaker's voice on input with speaker turns. However, proper training data is required to enable such models to learn so.

**ConcatAug** To enable direct S2ST models to preserve each speaker's voice for input with speaker turns, we augmented the training data by randomly sampling pairs of training examples and concatenating the source speech, the target speech, and the target phoneme sequences into new training examples. The result new examples contain two speakers' voices in both the source and the target speech, which enables the model to learn on examples with speaker turns. See Figure 2 for an example of such concatenation and the prediction from Translatotron 2 on it.

Such augmentation does not only enable the model to learn voice retention on speaker turns, but also increases the diversity of the speech content as well as the complexity of the acoustic conditions in the training examples, which can further improve the translation quality of the model, especially on small datasets (as shown in Section 5.1). Narayanan et al. (2019) uses a similar augmentation but in a more complicated fashion, for improving ASR performance on multi-speaker inputs.

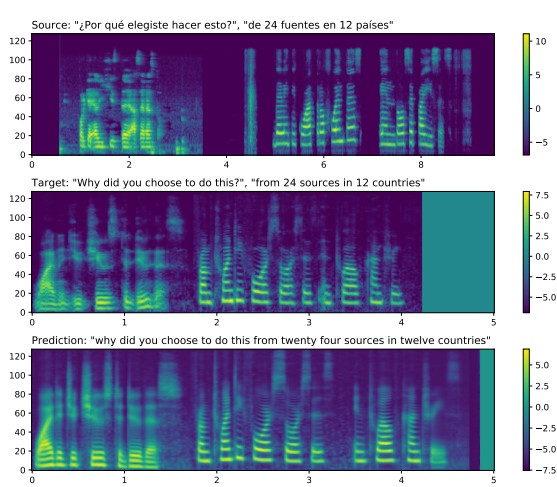

Figure 2: Sample mel-spectrograms on input with speaker turns. The input is a concatenation of an utterance from a male speaker followed by another utterance from a female speaker. Translatotron 2 preserves the voices of each speaker in the translated speech.

## 5 EXPERIMENTS

We conducted experiments on three datasets, including two Spanish→English datasets and a multilingual→English dataset. All datatsets use TTS synthesized target speech with 24 kHz sample rate. The phonemes used only at training time were converted from the transcripts using a proprietary G2P system. See Table 1 for the details of each dataset. We evaluated the translation quality, naturalness and robustness of the predicted speech, as well as speaker similarity for voice retention.

Unless described otherwise, in all the experiments, Translatotron 2 models use a 16-layer Conformer encoder with 144 dimension following Gulati et al. (2020), a 4-layer LSTM decoder, and a RNN-based synthesizer following Shen et al. (2020). A comprehensive table of hyperparameters in available in Appendix A. All models were implemented using the Lingvo framework (Shen et al., 2019).

### 5.1 TRANSLATION QUALITY

The same two datasets from Jia et al. (2019b) were used for evaluating the translation quality of Translatotron 2 when predicts translated speech is in a single female speaker's voice. In contrast to

Table 2: S2ST performance for outputting in a single speaker's voice. BLEU scores were computed with 1 reference for the Conversational test set, and with 4 references for the Fisher test set.

| | Conversational | | | Fisher | | |
|---|---|---|---|---|---|---|
| | BLEU | MOS | UDR (%) | BLEU | MOS | UDR (%) |
| Translatotron 2 | 55.6 | $4.21 \pm 0.06$ | 0.16 | 37.0 | $3.98 \pm 0.08$ | 0.07 |
| + ConcatAug | 55.1 | $4.19 \pm 0.06$ | 0.13 | 40.3 | $3.79 \pm 0.09$ | 0.14 |
| Translatotron | 50.4 | $4.15 \pm 0.07$ | 0.69 | 26.9 | $3.70 \pm 0.08$ | 0.48 |
| Cascade (ST → TTS) | 58.8 | $4.31 \pm 0.06$ | 0.21 | 43.3 | $4.04 \pm 0.08$ | 0.13 |
| Training target | 81.9 | $3.37 \pm 0.09$ | 0.43 | 88.6 | $3.95 \pm 0.07$ | 0.07 |
| *Reported in Jia et al. (2019b):* | | | | | | |
| Translatotron | 42.7 | $4.08 \pm 0.06$ | - | 25.6 | $3.69 \pm 0.07$ | - |
| Cascade (ST → TTS) | 48.7 | $4.32 \pm 0.05$ | - | 41.4 | $4.09 \pm 0.06$ | - |
| Training target | 74.7 | $3.71 \pm 0.06$ | - | 85.3 | $3.96 \pm 0.06$ | - |
| *Reported on discrete representation-based cascade systems:* | | | | | | |
| Zhang et al. (2021) (trained w/o text) | - | - | - | 9.4 | - | - |
| Lee et al. (2021a) (trained w/ text) | - | - | - | 37.2 | - | - |

Jia et al. (2019b), we did not augment the source speech to add background noise or reverberation, and we consistently use 80-channel mel-spectrogram as input and 128-channel mel-spectrogram as output for all Translatotron 2 models. The predicted mel-spectrogram is converted into waveform using the Griffin-Lim algorithm (Griffin & Lim, 1984).

Following Jia et al. (2019b), the translation quality is measured by BLEU on ASR transcribed text (lower case, no punctuation marks). Because ASR makes errors, such BLEU can be thought a lower bound of the translation quality. We used an ASR model from Park et al. (2020), trained on LibriSpeech (Panayotov et al., 2015) and LibriLight (Kahn et al., 2020) corpora. We retrained the baseline Translatotron strictly following Jia et al. (2019b) and re-evaluated it with this ASR model.

As shown in Table 2, the translation quality from Translatotron 2 outperformed the original Translatotron by a large margin and got very close to a strong cascade system. Applying ConcatAug further improved the translation quality of Translatotron 2 on the smaller Fisher dataset.

The original Translatotron uses the phoneme sequences from both the source and the target sides as auxiliary training tasks. Translatotron 2 uses only the target phonemes, yet yield better translation quality than Translatotron especially on the smaller Fisher dataset, indicating it is more data efficient.

Manual error case analysis revealed high consistency between the target speech prediction and the target phoneme prediction, suggesting more headroom for improving translation quality lies in the encoder and the decoder of Translatotron 2, instead of the synthesizer. Potential approaches to take include utilizing beam search, self-supervised pre-training (Baevski et al., 2020; Wang et al., 2021b), self-training (Park et al., 2020; Wang et al., 2021b), and weakly supervised data (Jia et al., 2019a).

## 5.2 SYNTHESIS NATURALNESS

The naturalness of the predicted speech is evaluated by subjective listening test, reporting 5-scale mean opinion scores (MOS) with 95% confidence interval on 1,000 randomly sampled predictions. A WaveRNN-based neural vocoder (Kalchbrenner et al., 2018) was used for converting the mel-spectrograms predicted from S2ST models into waveforms.

The result is reported in Table 2. The naturalness of the speech predicted from Translatotron 2 is significantly better than from the original Translatotron, and is getting close to a cascade system, which uses Tacotron 2, one of the state-of-the-art TTS models, to synthesize the predicted text translation into speech.

Consistent with Jia et al. (2019b), despite that the training targets in the Conversational dataset is synthesized with a lower quality Griffin-Lim vocoder, the trained S2ST model is able to synthesize with significantly better naturalness than the training target when a higher quality neural vocoder is used at inference time.

Table 3: S2ST performance with voice retention using the approach in Section 4. Speaker similarity MOS is evaluated between the synthesized English speech and the human Spanish recording. Note: 1) the BLEU evaluation set is a subset of the same in Table 2; 2) Training targets are human recordings in Jia et al. (2019b) and synthesized speech in this work.

|  | BLEU | Naturalness (MOS) | Similarity (MOS) |
|---|---|---|---|
| Translatotron 2 | 57.3 | $3.24 \pm 0.08$ | $2.33 \pm 0.08$ |
| + ConcatAug | 56.8 | $2.94 \pm 0.08$ | $2.12 \pm 0.07$ |
| Translatotron | 48.5 | $2.55 \pm 0.09$ | $2.30 \pm 0.07$ |
| + ConcatAug | 51.3 | $2.76 \pm 0.09$ | $2.19 \pm 0.07$ |
| Training target | 81.3 | $3.40 \pm 0.08$ | $2.55 \pm 0.07$ |
| *Results from Jia et al. (2019b):* |  |  |  |
| Translatotron | 36.2 | $3.15 \pm 0.08$ | $1.85 \pm 0.06$ |
| Training target | 59.9 | $4.10 \pm 0.06$ | - |

## 5.3 ROBUSTNESS

We specifically evaluated the robustness issue of over-generation in the predicted speech, such as babbling or long pause, measured by unaligned duration ratio (UDR) (Shen et al., 2020) with 1-second threshold.[1] The ASR transcribed text is used for alignment, using a confidence islands-based forced alignment model (Chiu et al., 2018).

The result is shown in Table 2. On the Fisher set, the UDR from Translatotron 2 is about 7 times lower than from the original Translatotron, and is about the same as the training target. On the Conversational set, the UDR from Translatotron 2 is more than 4 times lower than from the original Translatotron, and is even about 3 times lower than the training targets. Note that the training target in the Conversational set is synthesized by the Tacotron 2 TTS model (see Table 1), which by itself suffers from over-generation (He et al., 2019; Zheng et al., 2019; Battenberg et al., 2020; Shen et al., 2020). The result suggests that Translatotron 2 drastically improved robustness than the original Translatotron, and is also robust to a small ratio of disfluency in the training targets.

## 5.4 VOICE RETENTION

To evaluate the ability of retaining speakers' voices while translating their speech from one language to another, we augmented the Conversational dataset by synthesizing the target speech using a voice-transferring TTS model as described in Section 4.1. Examples with source speech shorter than 1 second were discarded for the stability of voice transferring. The result dataset contains parallel utterances with similar voices on both sides. S2ST models were trained on this dataset without any explicit conditioning on speaker embeddings or IDs (i.e. no speaker encoder for the original Translatotron). Translatotron 2 models used a Conformer encoder with a larger dimension (256) than in previous experiments, since its output was expected to carry more acoustic information for voice retention. Following Jia et al. (2019b), we also reduced the synthesizer's pre-net dimension to 16 to encourage it to infer voice information from the encoder output instead of the teacher-forcing label.

5-scale subjective MOS on both naturalenss and speaker similarity was evaluated with 1,000 random samples or pairs of samples from the test set[2], reported with 95% confidence interval. As Table 3 shows, both Translatotron 2 and Translatotron performed well using the new approach for voice retention. They both obtained about the same speaker similarity MOS as the original Translatotron and significantly better translation quality. Translatotron 2 further outperformed Translatotron in terms of translation quality, which is consistent with the experimental results for translating into a single speaker's voice. It is worth to note that with the new approach, the speaker similarity from S2ST models is capped by the same of the training targets, which by itself is low. This can be partially due to the performance of the voice-transferring TTS model used for synthesizing the training targets, and partially due to the fact that cross-lingual speaker similarity evaluation is more challenging to raters

---

[1]Under-generation (i.e. WDR from Shen et al. (2020)) does not apply because of the nature of translation. Related errors are reflected in the BLEU evaluation.

[2]This dataset was created by crowd-sourcing without collecting speaker identities for privacy protection. The test set (12.7 hours source recordings) may contain both speakers seen and unseen during training.

Table 4: Voice retention performance on speaker turns. The speaker similarity MOS between the leading/trailing 1.6-second segment from the predicted speech (English) and the entire 1st/2nd source speaker's speech (Spanish) is measured. ($\uparrow$ / $\downarrow$ indicates that higher/lower values are better.)

| | 1st source speaker | | 2nd source speaker | |
|---|---|---|---|---|
| | Leading seg. $\uparrow$ | Trailing seg. $\downarrow$ | Leading seg. $\downarrow$ | Trailing seg. $\uparrow$ |
| Translatotron 2 | $2.22 \pm 0.07$ | $2.15 \pm 0.07$ | $2.04 \pm 0.07$ | $2.00 \pm 0.07$ |
| + ConcatAug | $2.44 \pm 0.07$ | $1.82 \pm 0.07$ | $1.76 \pm 0.07$ | $2.51 \pm 0.08$ |
| Translatotron | $1.87 \pm 0.06$ | $1.90 \pm 0.07$ | $2.06 \pm 0.07$ | $2.05 \pm 0.07$ |
| + ConcatAug | $2.18 \pm 0.07$ | $1.71 \pm 0.06$ | $1.93 \pm 0.07$ | $2.35 \pm 0.07$ |
| Training target | $2.58 \pm 0.08$ | $1.62 \pm 0.06$ | $1.83 \pm 0.07$ | $2.44 \pm 0.07$ |

(some rating comments are purely based on language difference), as also observed in Zhang et al. (2019). Obtaining better quality training targets, such as human recordings instead of synthesized speech, may further improve the performance of S2ST models trained with the new approach.

### 5.4.1 SPEAKER TURNS

Speaker similarity evaluation with speaker turns on the entire model prediction is challenging because it would require speaker diarization on the predicted speech, and the potential content re-ordering during translation as well as potential model prediction error adds extra difficulty. We approximated by considering the leading/trailing short segments in the predicted speech as corresponding to each of the two speakers in inputs with a single speaker turn.

The evaluation set was constructed by applying the same concatenation augmentation as described in Section 4.2 on the original test set, so that each utterance contains two speakers' voices. Examples with target speech shorter than 2 seconds before concatenation were discarded. We conducted subjective speaker similarity MOS evaluation between the two entire original utterances before concatenation and the leading/trailing 1.6-second segments[3] of the predicted speech.

The results are shown in Table 4. It can be seen that the impact of the concatenation augmentation is consistent on Translatotron and Translatotron 2. When the concatenation augmentation was not used during training, for each source speaker, the similarity compared to the leading/trailing segment in the predicted speech was about the same; and for each segment in the predicted speech, the similarity compared to the first/second source speaker was also close. This suggests that the predicted speech was in a single speaker's voice, which was optimized for both source speakers at the same time. When the concatenation augmentation was used, both models obtained significantly higher speaker similarity on matched pairs than mismatched pairs, indicating that the models successfully separated two speakers and retained voice for each of them respectively. It can also be seen that Translatotron 2 obtained significantly higher speaker similarity than Translatotron on matched pairs, indicating the effectiveness of Translatotron 2.

Such quantitative evaluation cannot reflect how the predicted speech transits from one speaker's voice into another speaker's voice. Listening to the audio samples verified that the voice change happened instantly without blurry rather than transitionally, suggesting that the encoder outputs preserved the voice information locally in together with the linguistic information, and the synthesizer and the decoders were able to utilize such local information for translating the linguistic while retaining the corresponding voice information locally. A sample of such speaker turn with the prediction from Translatotron 2 trained with concatenation augmentation is visualized in Figure 2.

Although Translatotron 2 bears some similarity to cascade systems in terms of the cascade of the decoder and the synthesizer, such voice retention capacity, especially on speaker turns, is very challenging for the latter, as it would require separate speaker diarization and voice encoder.

While ConcatAug is effective on enabling S2ST models to support voice retention on speaker turns, and can further improve the translation quality and the speech naturalness on models with lower performance (e.g. trained on small datasets), it may negatively impact the speech naturalness and similarity on models with strong performance, as shown in Table 2 and Table 3. This may be

---

[3]1.6 seconds is the minimum duration for computing d-vectors for the objective evaluation in Appendix B.

Table 5: Multilingual X→En S2ST performance on 4 high-resource languages from CoVoST 2, measured by BLEU on ASR transcribed text. The same checkpoints from each model were used for evaluating all language pairs. Note: BLEU scores are not directly comparable between S2ST and ST.

| Source language | fr | de | es | ca |
|---|---|---|---|---|
| Translatotron 2 | 27.0 | 18.8 | 27.7 | 22.5 |
| Translatotron | 18.9 | 10.8 | 18.8 | 13.9 |
| ST (Wang et al., 2021a) | 27.0 | 18.9 | 28.0 | 23.9 |
| Training target | 82.1 | 86.0 | 85.1 | 89.3 |

explained by the fact that the augmented utterances sound less natural and supporting speaker turns may sacrifice model capacity on single-speaker cases.

## 5.5 MULTILINGUAL S2ST

We also conducted experiments to evaluate the performance of multilingual X→En S2ST models. We trained both a Translatotron 2 model and a Translatotron model on the 4 high-resource languages from the CoVoST 2 corpus (Wang et al., 2021a), using TTS synthesized target speech in a single female speaker's voice. The original Common Voice (Ardila et al., 2020) data split was followed. The models were not explicitly conditioned on language IDs. We modified the original Translatotron model to use the same Conformer encoder and SpecAugment as in Translatotron 2 for a fair comparison. Similarly, we only used the target phoneme sequence (in English) for both Translatotron and Translatotron 2. Translatotron used a decoder with 6-layer LSTM with 1024 dimension; Translatotron 2 used a decoder with 6-layer LSTM with 512 dimension and a synthesizer of 2-layer LSTM with 1024 dimension. Both used the same convoluational post-net as Shen et al. (2020). The total number of parameters in Translatotron 2 was about 10% fewer than in Translatotron.

The translation quality as measured by BLEU on the ASR transcribed text is shown in Table 5. The BLEU scores from Translatotron 2 significantly outperformed the same from Translatotron. Although the results are not directly comparable between S2ST and ST,[4] the close numbers suggest that Translatotron 2 achieved translation quality comparable to the baseline ST model. This indicates that Translatotron 2 is also highly effective for multilingual S2ST.

## 6 CONCLUSION

We proposed *Translatotron 2*, a neural direct S2ST model that can be trained end-to-end. The major differences compared to the original Translatotron are: 1) the output from the auxiliary target phoneme decoder is used as an input to the spectrogram synthesizer; 2) the spectrogram synthesizer is duration-based, while still keeping the benefits of the attention mechanism. Experiments conducted on three different datasets, including multilingual S2ST, suggested that Translatotron 2 outperformed the original Translatotron by a large margin in terms of translation quality and predicted speech naturalness, and drastically improved the robustness of the predicted speech.

We also proposed a new method for retaining the source speaker's voice in the translated speech. In contrast to the original Translatotron, S2ST models trained with the new method is restricted to retain the source speaker's voice, but not able to generate speech in a different speaker's voice, which makes the model free from potential abuse such as creating spoofing audios, thus more robust for production deployment. When the new method is used together with a simple concatenation data augmentation, the trained Translatotron 2 model is able to retain each speaker's voice for input with speaker turns.

ETHICS STATEMENT

Preserving voices during S2ST is desired as it helps communication. However, progress on high quality voice cloning poses societal concerns on misuse, such as creating "deepfake" spoofing audios.

---

[4]BLEU for S2ST is computed case-insensitively and without punctuation marks because the transcript is from the output of ASR. On the other side, ASR errors in evaluation underestimates the performance of S2ST.

This work includes a new voice retention method for S2ST that reduces the potential for such misuse, if such models are deployed in production.

## REPRODUCIBILITY STATEMENT

We conducted experiments on multiple datasets, including two public datasets. We also provided a comprehensive list of hyperparameters in Appendix A to help reproducing the experimental results.

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

## A    TABLE OF HYPER-PARAMETERS

Table 6: Model hyper-parameters used in the experiments ($\times n$ means $n$ layers).

| | Fisher | CoVoST 2 | Conversational (Single voice) | Conversational (Voice retained) |
|---|---|---|---|---|
| *Input* | | | | |
| Sample rate (Hz) | 8,000 | 48,000 | 16,000 - 48,000 | |
| Mel channels | | | 80 | |
| Mel lower band (Hz) | | | 125 | |
| Mel upper band (Hz) | 3,800 | 7,600 | 7,600 | |
| Frame size (ms) | | | 25 | |
| Frame step (ms) | | | 10 | |
| *Output* | | | | |
| Sample rate (Hz) | | | 24,000 | |
| Mel channels | | | 128 | |
| Mel lower band (Hz) | | | 20 | |
| Mel upper band (Hz) | | | 12,000 | |
| Frame size (ms) | | | 50.0 | |
| Frame step (ms) | | | 12.5 | |
| *SpecAugment* | | | | |
| Freq blocks | | | 2 | |
| Time blocks | | | 10 | |
| Freq block max length ratio | | | 0.33 | |
| Time block max length ratio | | | 0.05 | |
| *Encoder* | | | | |
| Conformer dims | $144 \times 12$ | $144 \times 16$ | $144 \times 16$ | $256 \times 16$ |
| Attention heads | 4 | 4 | 4 | 4 |
| Conv kernal size | 32 | 32 | 32 | 32 |
| Subsample factor | 4 | 4 | 4 | 4 |
| *Attention* | | | | |
| Output dim | 256 | 512 | 512 | 512 |
| Hidden dim | 512 | 512 | 512 | 512 |
| Attention heads | 4 | 8 | 8 | 8 |
| Dropout prob | 0.1 | 0.2 | 0.2 | 0.2 |
| *Decoder* | | | | |
| LSTM dims | $256 \times 4$ | $512 \times 6$ | $512 \times 4$ | $512 \times 4$ |
| Zoneout prob | 0.1 | 0.1 | 0.1 | 0.1 |
| Phoneme embedding dim | 96 | 256 | 256 | 256 |
| Label smoothing uncertainty | 0.1 | 0.1 | 0.1 | 0.1 |
| Loss weight | 1.0 | 1.0 | 1.0 | 1.0 |
| *Duration predictor* | | | | |
| Bi-LSTM (dim $\times$ layers) | $64 \times 2$ | $128 \times 2$ | $128 \times 2$ | $128 \times 2$ |
| Loss weight | 1.0 | 1.0 | 1.0 | 1.0 |
| *Synthesizer* | | | | |
| LSTM dims | | | $1024 \times 2$ | |
| LSTM zoneout prob | | | 0.1 | |
| Pre-net dims | $128 \times 2$ | $128 \times 2$ | $128 \times 2$ | $16 \times 2$ |
| Pre-net dropout prob | | | 0.5 | |
| Post-net (kernel, channels) $\times$ layers | | | $(5, 512) \times 4 + (5, 128)$ | |
| Loss weight | | | 0.1 | |
| *Training* | | | | |
| Batch size | 1,024 | 768 | 768 | 768 |
| $L^2$ regularization weight | $10^{-6}$ | $10^{-6}$ | $10^{-6}$ | $10^{-6}$ |
| Learning rate (Transformer schedule) | 5.0 | 3.75 | 4.0 | 4.0 |
| Warm-up steps (Transformer schedule) | 10K | 20K | 10K | 10K |
| Training steps | 120K | 130K | 220K | 150K |

Figure 3: Affinity matrices of d-vector similarity among 100 random examples. ("s2st" refers to the predictions from Translatotron 2.)

Table 7: Objective d-vector similarity between the predicted translated speech (English) and the source human speech (Spanish) on speaker turns. The similarity between the leading/trailing 1.6-second segment from the predicted speech and the entire 1st/2nd source speaker's speech is measured. ($\uparrow$ / $\downarrow$ means higher/lower values are better.)

|  | 1st source speaker | | 2nd source speaker | |
|---|---|---|---|---|
|  | Leading seg. $\uparrow$ | Trailing seg. $\downarrow$ | Leading seg. $\downarrow$ | Trailing seg. $\uparrow$ |
| Translatotron 2 | 0.21 | 0.19 | 0.21 | 0.19 |
| + Concat aug. | 0.20 | 0.14 | 0.14 | 0.21 |
| Translatotron | 0.20 | 0.22 | 0.27 | 0.29 |
| + Concat aug. | 0.32 | 0.16 | 0.14 | 0.35 |
| Training target | 0.48 | 0.17 | 0.15 | 0.48 |

## B  OBJECTIVE SPEAKER SIMILARITY ANALYSIS

Subjective speaker similarity evaluation is costly and has long turnaround. We explored into alternative objective evaluation using separately trained speaker encoders, such as d-vector (Wan et al., 2018). We evaluated the voice retention performance using the cosine similarity of the d-vectors.

We first checked the scenario that each input contains a single speaker's recording. Figure 3 visualizes the affinity matrices of d-vector similarity among different input utterances for a Translatotron 2 model. The outstanding higher similarity values on the diagonals indicate that the model is able to retain the source speaker's voice in the predicted speech.

We then conducted a detailed evaluation for the voice retention performance for speaker turns. The experiment setting up was identical to Section 5.4.1, except that the speaker similarity was measured by d-vector similarity instead of subjective MOS evaluation. The d-vectors for each source speaker were computed on the entire original utterance before concatenation; the d-vectors for each speaker in the prediction is approximated by computing on the leading/trailing 1.6 seconds of predicted speech.

The results are shown in Table 7. Consistent with the MOS evaluation results in Table 4, when the concatenation augmentation was not used, the d-vector similarity to each source speaker is about the same regardless if it was compared to the leading or trailing segments, indicating that the predicted speech was in a single speaker's voice and the model was unable to separate different speakers in the input, but rather optimized for both source speakers at the same time. When the concatenation augmentation was used, the d-vector similarity was significantly higher between matched pairs than between unmatched pairs, indicating that the models were able to separate different speakers in the input and retain their voice in the predicted speech respectively.

However, when these similarities are compared among different models, it seems suggesting that Translatotron performed better than Translatotron 2, which is in contradictory to the subjective evaluation results in Table 4. By carefully listening to the audio samples, we found that such discrepancy may be due to that the d-vector model was also sensitive to non-voice related acoustic characteristics, such as reverb and channel noise in the audios. This is likely a consequence of the fact that in the large-scale training set for the d-vector model used in the evaluation, each speaker is typically associated with a particular recording condition, e.g. recording device and room. Because the encoder output from the Translatotron model was of significantly larger dimension than from the

Table 8: Translatotron 2 performance on the Conversational dataset using an autoregressive synthesizer and a non-autoregressive synthesizer.

| Synthesizer | BLEU | MOS |
|---|---|---|
| RNN | 55.6 | $4.21 \pm 0.06$ |
| Conformer | 54.5 | $3.61 \pm 0.09$ |

Table 9: Ablation studies on the CoVoST 2 dataset (on 4 high-resource X→En pairs). $+ / -$ indicates using or replacing a component.

| Source language | fr | de | es | ca |
|---|---|---|---|---|
| Translatotron (w/ SpecAugment) | 17.7 | 9.9 | 17.7 | 13.1 |
| + Conformer encoder | 18.9 | 10.8 | 18.8 | 13.9 |
| + NAT decoder | 4.0 | 2.1 | 3.5 | 2.5 |
| Translatotron 2 | 27.0 | 18.8 | 27.7 | 22.5 |
| − Conformer encoder | 26.4 | 18.1 | 26.4 | 21.8 |
| − NAT synthesizer | 26.9 | 18.3 | 27.0 | 22.0 |
| − SpecAugment | 25.9 | 17.9 | 25.9 | 21.8 |
| Training target | 82.1 | 86.0 | 85.1 | 89.3 |

Translatotron 2 model (2048 vs 256), it was capable of carrying more non-voice acoustic information and thus obtained better d-vector similarity, which not necessarily indicating higher speaker similarity.

These results suggest that while such speaker encoder-based objective analysis reveals insightful indications about the performance of the S2ST models, it can be less reliable compared to subjective MOS evaluation. Such reliability also highly depends on the training details of the speaker encoder model being used, especially the training corpus.

## C  NON-AUTOREGRESSIVE SYNTHESIZER

Following recent non-autoregressive TTS works (Ren et al., 2021; Guo et al., 2021; Lee et al., 2021b; Elias et al., 2021a), we explored using non-autoregressive synthesizer in Translatotron 2, which may have significantly lower latency at inference time. The experimental results in Table 8 suggested that despite producing comparable BLEU on ASR transcribed text from the Translatotron 2 predictions, using non-autoregressive synthesizer produced significantly worse naturalness of the predicted speech. This is consistent with the observation in TTS in Shen et al. (2020).

## D  ABLATION STUDIES

To understand the importance of each components in the Translatotron 2 model, we conducted ablation studies on the CoVoST 2 multilingual X → En dataset as described in Section 5.5. All models in the ablation used the same input and output features, SpecAugment setting, and learning rate schedule as described in Section 5.5 and Appendix A. No auxiliary training target on the source text or phonemes were used. For models using an RNN-based encoder, we first applied the same 4× time-wise subsampling as used in the Conformer encoder, then used a 8-layer bidirectional LSTM stack with a cell size of 256. The number of parameters in this RNN-based encoder is close to the same in the Conformer encoder. For the Translatotron model using a NAT decoder, the same architecture and hyperparameters as the synthesizer of Translatotron 2 was used to replace the original attention-based decoder. For Translatotron 2 using a non-autoregressive synthesizer, a simple 6-layer Conformer stack with a dimension of 512 and 8 attention heads was used to replace the autoregressive synthesizer, same as in Appendix C. This Conformer-based non-autoregressive synthesizer is similar to the Transformer-based decoder in the FastSpeech 2 TTS model (Ren et al., 2021), but performed better in our experiments. All the rest hyperparameters follow Appendix A for Translatotron 2, and follow the Conversational model in Jia et al. (2019b) for Translatotron. All models were trained for

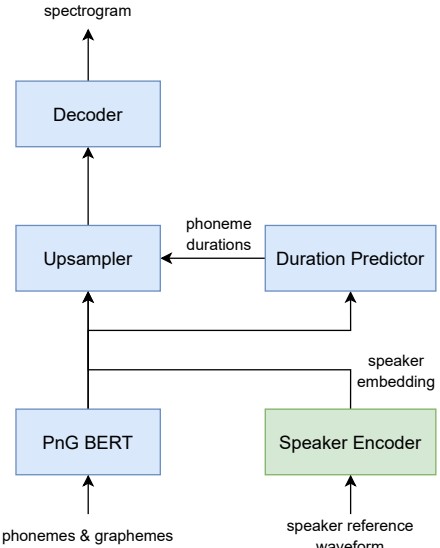

Figure 4: Augmented PnG NAT TTS model for cross-lingual voice transferring.

200K steps with a batch size of 768. The evaluation was done using the same checkpoints for all the 4 language pairs, picked by the highest average performance on the dev sets.

The results are shown in Table 9. As can be seen, while the use of Conformer, SpecAugment, and NAT decoder helps the performance of Translatotron 2, replacing them with alternative architectural choices or removing SpecAugment only reduced the performance by a small degree ($< 2$ BLEU). Similarly, directly using these components in the original Translatotron does not bring its performance close to Translatotron 2. These results suggest that the performance improvement of Translatotron 2 comes from addressing the weaknesses existing in Translatotron (Section 3), rather than the architectural choices of each individual components.

Consistent with the result in Appendix C, using a non-autoregressive synthesizer in Translatotron 2 obtained comparable translation quality to using an autoregressive synthesizer which is based on the NAT decoder. However, as Appendix C shows, such non-autoregressive synthesizer may suffer from lower naturalness in the predicted speech. Directly using the NAT decoder in the original Translatotron obtained the worst performance because its monotonic nature does not work well for a translation task.

## E   CROSS-LINGUAL VOICE TRANSFERRING TTS

The TTS model used for synthesizing training targets in the source speaker's voice in Section 4.1 is modified from the PnG NAT (Jia et al., 2021) TTS model by incorporating a separately trained speaker encoder (Wan et al., 2018) in the same way as Jia et al. (2018). The architecture of this TTS model is illustrated in Figure 4. We trained this model on the LibriTTS corpus (Zen et al., 2019), following the hyperparameters in Jia et al. (2021); Shen et al. (2020); Wan et al. (2018). The speaker encoder is separately trained in a speaker verification task and is frozen during the TTS model training. At synthesis time, the phonemes and graphemes in the target language and the reference speech in the source language are fed into the model as inputs; and the model produces speech in the target language with the voice from the source speech transferred.

