# OpenReview forum: "Translatotron 2: Robust direct speech-to-speech translation"
_ICLR.cc/2022/Conference — ICLR 2022 Submitted_

### Official Review · Reviewer_gzba · 2021-10-28

**Correctness:** 3
**Technical Novelty And Significance:** 2
**Empirical Novelty And Significance:** 2
**Recommendation:** 6
**Confidence:** 4

**Main Review:**

Please list both the strengths and weaknesses of the paper. When discussing weaknesses, please provide concrete, actionable feedback on the paper.

This paper describes an updated version of Translatron which purports to be a speech to speech translation system trained end-to-end.  The claimed benefits include the ability to use with languages without a written form and reduced error-compounding relative to a cascade system.  However, I find no support for these claims in the paper.  The system is not strictly end-to-end since it relies on an auxiliary loss function based on a phoneme transcription and nowhere does it explain where this transcription comes from (the original Translatotron paper is also quiet about this).   Since the experimental data was derived from text corpora my guess is that the phoneme transcription used in the experiments was generated by a standard G2P system.  Given the difficulty of automatic phone transcription, its not at all clear how you could in practice realise the claimed benefit of using with languages with only speech as training data and no text (or well-trained ASR system)

The results presented for translation quality come close to those achieved by an ST+TTS cascade system but do not match it.  Given that deriving an accurate phoneme transcription is not much easier than deriving a text transcription, I fail to see the benefits of this system.

Also the so-called Conversational dataset is mis-leading because this is actually crowd-sourced speakers reading conversational texts.  Read speech is quite different from spontaneous conversational speech.

The architectural changes between Translatotron and Translatotron 2 are described but not fully justified.  The paper contains various evaluations of voice quality and similarity but does not provide any ablation studies to show why the new architecture is better than the old and the relative contribution of each change.

One particular difficulty is understanding the BLEU results since this is measured automatically by applying ASR to the output.  I presume the reason why improving the output synthesis quality improves the BLEU score is not because the underlying translation is any better, but simply because the ASR made fewer errors?

The authors include an augmentation option in training to include more than one speaker turn with the goal of getting the target speech to track the changed input speech instead of being an average of the two.  The results show that without the augmentation, the target voices are indistinguishable, whereas with it there is a clear differentiation although the absolute similarities are quite poor.

One of the many difficulties that i had in understanding this paper was the lack detail on the training data and how the results would apply to a real-world use-case.  Firstly, I cant find any information on the relative sizes of the train and test sets.  Are speakers in the test set included in the training set - ie is this evaluated as a speaker independent translation system or a speaker dependent system.  For the speaker dependent case,  if I wanted to train a Translatotron for a single voice, how much speech data would I need?

Overall, there is much interesting stuff in this paper, but far too many unanswered questions.


**Summary Of The Paper:**

This paper describes an updated version of Translatron which purports to be a speech to speech translation system trained end-to-end.  The new system now includes the ability to control the target voice to match the input voice without providing an explicit speaker id vector and the translation performance of the updated system now comes close to that of a cascade speech to text + TTS system, although it does not match or improve on it.


**Summary Of The Review:**

The paper describes an incremental improvement over a previous paper on the same topic.  However, the experimental details are not adequately described and the claimed benefits relative to a cascade system are not justified.

I remain somewhat sceptical of the practical utility of the overall framework.  Nevertheless, following the authors response to questions and updated paper, I have increased my score to "marginally above" the threshold.

---

> ### Author Response · Authors · 2021-11-16
> **Responses**
>
> We thank the reviewer for the helpful feedback. Below are the itemized responses regarding each comment. We have also incorporated these into the revised version.
>
> **1. Re: The claimed benefits include the ability to use with languages without a written form and reduced error-compounding relative to a cascade system. However, I find no support for these claims in the paper.**
>
> We did not make such claims. The relevant discussion in Section 1 paragraph 2 is a general discussion on the potential benefits of direct S2ST, with references to existing works. Not all such potential benefits are realized by every direct S2ST model.
>
> **2. Re: The system is not strictly end-to-end since it relies on an auxiliary loss function based on a phoneme transcription and nowhere does it explain where this transcription comes from.**
>
> Using auxiliary loss does not discount the model being end-to-end. Similar examples include many widely accepted object detection works such as Fast R-CNN (Girshick, 2015) and YOLO (Redmon et al., 2016).
>
> We explained in Section 5, Section 4, and Table 1 that the target speech in the training sets were synthesized by TTS on target transcripts. We revised Section 5 to clarify that the phonemes are generated by a G2P system from the target transcripts.
>
> **3. Re: Given that deriving an accurate phoneme transcription is not much easier than deriving a text transcription, I fail to see the benefits of this system.**
>
> Direct benefits of this system include preserving the speaker’s voice during S2ST, even on speaker turns, which is challenging to achieve with cascade approaches, despite of the availability of phonemes.
>
> Besides that, we believe that there are broader impact of this work for the possibilities it enabled for further works (e.g. simultaneous S2ST, prosody transferring in S2ST, etc.).
>
> **4. Re: So-called Conversational dataset is mis-leading because this is actually crowd-sourced speakers reading conversational texts. Read speech is quite different from spontaneous conversational speech.**
>
> We clearly stated in Table 1 that this dataset is read conversational speech.
>
> We considered renaming this dataset, but that would cause unnecessary confusion when comparing with previous works such as the original Translatotron. We welcome reviewer’s suggestions on how to improve.
>
> **5. Re: The architectural changes between Translatotron and Translatotron 2 are described but not fully justified. The paper contains various evaluations of voice quality and similarity but does not provide any ablation studies to show why the new architecture is better than the old and the relative contribution of each change.**
>
> We revised and added motivation for the architecture design of Translatotron 2 at the beginning Section 3, as well as the weaknesses of the original Translatotron that we addressed. We also added ablation studies in Appendix D.
>
> **6. Re: One particular difficulty is understanding the BLEU results since this is measured automatically by applying ASR to the output. I presume the reason why improving the output synthesis quality improves the BLEU score is not because the underlying translation is any better, but simply because the ASR made fewer errors?**
>
> Quality evaluation in an open challenge for S2ST. Measuring BLEU on ASR transcribed text is an established approach widely adopted in this domain (Salesky et al., 2021).
>
> BLEU increase cannot be explained by the MOS increase. Example data points include the first three rows on the Fisher dataset in Table 2. We also revised to add analysis on the translation quality in Section 5.1.
>
> **7. Re: The results show that without the augmentation, the target voices are indistinguishable, whereas with it there is a clear differentiation although the absolute similarities are quite poor.**
>
> The 5-point MOS ranges from 1.0 to 5.0 (1.0 means completely different). In Table 4 row 2, the relative difference between 1.76 and 2.51 is 99%, which is huge.
>
> **8. Re: One of the many difficulties that i had in understanding this paper was the lack detail on the training data and how the results would apply to a real-world use-case. Firstly, I cant find any information on the relative sizes of the train and test sets. Are speakers in the test set included in the training set.**
>
> The sizes of the training sets are reported in Table 1. We revised Section 5.4 and added the size of the test set.
>
> Unfortunately, this dataset (Conversational) is a crowd-sourced dataset without collecting speaker IDs due to privacy concerns. It’s reasonable to presume that there are both seen and unseen speakers in the test set, but it’s difficult to quantify. We revised Section 5.4 to clarify.
>
> **References:**
> - Girshick, Fast R-CNN, ICCV 2015.
> - Redmon et al., You only look once: Unified, real-time object detection, CVPR 2016.
> - Salesky et al., Assessing Evaluation Metrics for Speech-to-Speech Translation, ASRU 2021.

---

> > ### Comment · Reviewer_gzba · 2021-11-23
> > **Uprated review**
> >
> > I remain somewhat sceptical of the practical utility of the overall framework.  Nevertheless, following the authors response to questions and updated paper, I have increased my score to "marginally above" the threshold.

---

### Official Review · Reviewer_rKi2 · 2021-11-02

**Correctness:** 4
**Technical Novelty And Significance:** 3
**Empirical Novelty And Significance:** 2
**Recommendation:** 5
**Confidence:** 3

**Main Review:**

Strength:

1) With the use of Conformer encoder with Spec Augment and giving auxiliary target phoneme decoder output as input to the spectrogram synthesizer , the results in Table2 indicate the significant improvement in transcription quality

2) The voice retention capability of Translatotron 2 is significant

3) The ability of Translatotron2 to retain voice in case of speaker turns without additional diarization systems is interesting

4) The multilingual capability of Translatotron 2 is significant

Weakness:

1) While results are impressive, however, novelty is limited. For example: a) Conformer encoder and SpecAug are known to improve ASR performance; b) Using auxiliary target phoneme decoder output as input to the spectrogram synthesizer makes the architecture similar to cascade system; c) Robustness issue of Translatotron is addressed here by using Non-Attentive Tacotron spectrogram synthesizer

2) In both Table 2 and Table 3,  two results for the Translatotron model are shared, one of which is reported from the original paper. For the second results that are reported, it appears that the authors trained their own Translatotron model for the experiments, however, this should be stated explicitly in the paper

3) In Section 4.2, pause duration used to concatenate two consecutive speakers is missing in the description of  ConcatAug method. This information will be useful to replicate the experiments. Besides, it will justify the rationale of using 1.6 sec leading and trailing segments to measure voice retention in speaker turn experiments.


**Summary Of The Paper:**

Translatotron2, a speech-to-speech neural based translation system is proposed. The work is a modification of Translatotron (also a speech-to-speech translation system) and tries to address some of the  issues in Translatotron:

a) Translation quality fairly below a cascaded baseline

b) synthesized translated speech suffers from robustness issues, such as babbling and long pauses

c) Voice retention relies on explicit speaker embedding which can potentially be misused for generating spoofing audio with arbitrary content

Compared to Translatotron, following modifications are made in Translatotron2. The experiments suggest that Translatotron 2 significantly outperforms Translatotron, and is comparable to a cascaded system, in terms of translation quality, speech naturalness and speech robustness.

a)  Output from the auxiliary target phoneme decoder is used as an input to the spectrogram synthesizer

b) Conformer Encoder with SpecAugment is used, which is known to improve the speech to text performance

c) Duration-based  Spectrogram synthesizer is used. Specifically, architecture and hyperparameters similar to Non-Attentive Tacotron spectrogram synthesizer is used, which is known to improve the robustness issue in Tacotron2

d) To retain speakers’ voices across translation, Translatotron2 is trained on parallel utterances with the same speaker’s voice on both sides. This enables the trained model to restrict generation to only the source speaker’s voice, mitigating the risk of potential misuse for creating spoofing audio artifacts. Parallel utterances are synthesized using a TTS model with cross lingual voice transfer capacity.

In addition, authors also trained Translatotron 2 with examples that contain two speakers’ voices in both the source and the target. The experiments suggest that Translatotron 2 has capability to retain voice identity when the input contains speaker turns.  For multilingual experiments, Translatotron 2 was trained with 4 high resource languages  and BLEU scores suggest Translatotron2 outperforms Translatotron in multilingual settings as well.



**Summary Of The Review:**

The paper is well written.  The experiment results of  improvement in  translation quality , naturalness and robustness are convincing. The voice retention capability in the speaker turn setting of Translatotron 2 is appealing.  However, the overall  paper’s novelty is limited mainly because the paper brings in the best systems available in literature together to improve speech-to-speech translation systems performance. Thus, the paper might be more appropriate for a speech technology focused conference.

---

> ### Author Response · Authors · 2021-11-16
> **Responses**
>
> We thank the reviewer for the helpful feedback. Below are the itemized responses regarding each comment. We have also incorporated these into the revised version.
>
> **1. Re:  The overall paper’s novelty is limited mainly because the paper brings in the best systems available in literature together to improve speech-to-speech translation systems performance.**
>
>  - We added ablation studies in Appendix D. The results showed that simply bringing Conformer, SpecAugment, and NAT (Non-Attentive Tacotron) spectrogram decoder to the original Translatotron does not bring its performance close to Translatotron 2. Instead, the large performance improvement comes from the novel high-level architecture of Translatotron 2 (i.e. Figure 1 (a)). These results endorse the novelty of our model architecture contributions. We also revised Sec. 3 and added discussion on the weaknesses of the original Translatotron which we addressed in Translatotron 2.
>  - While the proposed model architecture is inspired from cascade systems, its achievement on voice retention (especially for the speaker turn case) is infeasible with typical cascade systems. Also as a comparison, Kano et al., 2021 is a more straightforward approach of direct S2ST simulating cascade systems, which unfortunately has to be trained in a complicated way and does not support voice retention.
>  - This work is the first direct S2ST model that achieved comparable performance (both translation quality and speech quality) to cascade systems (on 3 datasets, including both bilingual and multilingual). Furthermore, it’s the first S2ST model that is able to preserve each speaker’s voice on speaker turns. We believe that such results are important breakthroughs and will encourage the community on the research on direct S2ST, which is a research topic rarely touched in the past. We believe that these results justify strong empirical contributions.
>
> **2. Re: Table 2 and Table 3, two results for the Translatotron model are shared, one of which is reported from the original paper. For the second results that are reported, it appears that the authors trained their own Translatotron model for the experiments, however, this should be stated explicitly in the paper**
>
> We revised Section 5.1 to clarify.
>
> **3. Re: Pause duration used to concatenate two consecutive speakers is missing in the description of ConcatAug method.**
>
> We do not add extra pause duration when concatenating two utterances in ConcatAug.
>
> **4. Re: The rationale of using 1.6 sec leading and trailing segments to measure voice retention in speaker turn experiments.**
>
> We revised Section 5.4.1 to clarify. 1.6 second is the minimal duration required by the d-vector model, which we utilized in the objective evaluation for voice retention on speaker turns (Appendix B).

---

### Official Review · Reviewer_aQuX · 2021-11-03

**Correctness:** 3
**Technical Novelty And Significance:** 3
**Empirical Novelty And Significance:** 3
**Recommendation:** 6
**Confidence:** 2

**Main Review:**

The paper proposed several non-trivial improvement to a previous S2ST model, which significantly improves translation quality and speech naturalness, as well as mitigates audio spoofing concerns in production. Although some improvements are apparently inspired by existing work, they can still be considered somewhat novel and significant contribution.

Most of the claimed improvements are substantiated by experiments. However, some new questions arise from the experiment results themselves.
1. The reason to translation quality improvement (5.1, 5.5) is not discussed, and is not clear from the proposed approaches. Some ablation study would be helpful here.
2. Voice retention in speaker turns (5.4.1) seems somewhat tailored towards ConcatAug, and its relevance to real-world S2ST application is not clear. (How often will a third person use S2ST to translate dialogue between two other persons?) Instead it would be more interesting to see experiment scenarios related to anti-spoofing as discussed at the beginning of the paper.

The paper is generally written clearly and is easy to understand. But some important technical details are completely left out as reference to previous work, such as 3.3 and 4.1. It makes readers not closely following latest research in speech quite difficult to understand. In other words, the paper is not self-contained. In addition, requiring reviewers to search for these references during a double-blind review can easily reveal authors' identity or affiliation.

**Summary Of The Paper:**

The paper proposed a speech-to-speech translation (S2ST) model that is and improvement to a previous work. The model is trained end-to-end from speech to speech, along with an auxiliary speech-to-phoneme task. The relevance of the two tasks is further exploited beyond parameter sharing, by feeding phoneme decoder's hidden layer output to spectrogram synthesizer's input, and the synthesizer is duration based. To retain speakers' voice from source language to target language, two data-centric approach is proposed. First, a zero-shot voice-transfer TTS model is trained, to transfer target speech into source speaker's voice. Second, to enforce local voice similarity within an utterance, two samples from two distinct speakers are randomly selected and concatenated to create new training samples. Experiments are conducted on speech-to-text translation (ST) datasets with TTS synthesized target speech, and are measured by both objective (BLEU) and subjective (MOS) metrics. The proposed model's translation quality is much closer to the cascaded ST+TTS oracle than baseline S2ST models, for both bilingual and multilingual S2ST tasks. Its generated speech is also more natural and more resembles the source speech.

**Summary Of The Review:**

Contribution and novelty are moderately significant. Experiments firmly supports multiple claimed improvement to the previous model. Key technique are completely left out as reference to other work, making out-of-domain readers difficult to follow.

---

> ### Author Response · Authors · 2021-11-16
> **Responses**
>
> We thank the reviewer for the helpful feedback. Below are the itemized responses regarding each comment. We have also incorporated these into the revised version.
>
> **1. Re: The reason to translation quality improvement (5.1, 5.5) is not discussed, and is not clear from the proposed approaches. Some ablation study would be helpful here.**
>
> We revised and added ablation studies in Appendix D. We also added a discussion on the motivation of the architecture design of Translatotron 2 as well as the weaknesses of the original of Translatotron being addressed at the beginning of Section 3. We believe these added content well addressed this question.
>
> **2. Re: Voice retention in speaker turns seems somewhat tailored towards ConcatAug, and its relevance to real-world S2ST application is not clear. (How often will a third person use S2ST to translate dialogue between two other persons?)**
>
> Although we demonstrated this capacity with ConcatAug, we believe that such voice retention on speaker turns can be very valuable in the real world. Example scenarios include: a) speech translation for a multi-person video meeting; b) dubbing for a movie; etc.
>
> **3. Re: some important technical details are completely left out as reference to previous work, such as 3.3 and 4.1.**
>
> We revised Section 3.3 to add more details. We also added Appendix E to explain the TTS model architecture used in Section 4.1.

---

### Official Review · Reviewer_YR5V · 2021-11-07

**Correctness:** 4
**Technical Novelty And Significance:** 2
**Empirical Novelty And Significance:** 3
**Recommendation:** 5
**Confidence:** 5

**Main Review:**

The paper gives a good picture of current developments in speech synthesis and voice conversion, especially those that are derived from the Tacotron family of models. The setup seems to be directly derived (at least in philosophy) from the Translatotron. In Translatotron, the main components were an encoder and two types of decoders - one to synthetize target spectrogram and the other to classify phonemes in source/target languages. The language decoders in Translatotron were treated in the manner of multi task learning - to add extra supervision to the setup rather than to be used directly for the task. In the current work, the language decoders seem to be performing a similar role, but the paper does not go too much in detail about the philosophical motivations.

The paper is generally very well written and results (also, samples) look good. However, I have some misgivings about

1) The novelty of concepts. It seems that most of the ideas are already there in Translatotron, and this paper modifies architecture (and does not give much insight into these changes).
2) I did not understand the presentation in section 4.1 - of retaining voice. The paper proposes an alternative way to produce source speaker's voice without conditioning with a speaker embedding, which they claim could be used for spoofing purposes. They also point to other works and especially, state that they can produce target voice with the help of a speaker encoder network and TTS.

"We modified the PnG NAT (Jia et al., 2021) TTS model by incorporating a separately trained speaker
encoder (Wan et al., 2018) in the same way as Jia et al. (2018), and trained it on the LibriTTS corpus
(Zen et al., 2019). ..."

I find that this could be explained in a clearer manner. Do they convert to text and then to speech? Moreover, the use of a speaker encoder seems to suggest that the embedding will add on to the context as in other works, but then wasn't the aim of the work to avoid using a speaker encoder for antispoofing purposes? Please clarify.

Overall, I feel that the current work does not have sufficient novelty, and details are missing.

**Summary Of The Paper:**

This paper proposes a setup to perform Direct Speech To Speech Translation from one language to another. As such, we could view this as a voice conversion setup, but with the additional task of rendering translated voice in a different language. It seems to be created on the same lines as Translatotron, but with improved performance from different modeling choices. The architecture is encoder-decoder, with the decoder side consisting of two parts - a spectrogram synthesizer, and a phoneme/language translator. The phoneme translator is needed because it also needs to translate to a different language.

The experimental evaluation is quite good, consisting of cases to translate with the same voice as source (voice retention); samples with speaker turns (augmented dataset with two different speaker utterances concatenated together), and cross language translation. The paper shows improved results over Translatotron in all cases.

Aside from the results, the contributions seem to be incremental. They also emphasize a modified training setup that does not use speaker embeddings to avoid antispoofing.

**Summary Of The Review:**

Not enough novelty.
Some components not explained clearly,

---

> ### Author Response · Authors · 2021-11-16
> **Responses**
>
> We thank the reviewer for the helpful feedback. Below are the itemized responses regarding each comment. We have also incorporated these into the revised version.
>
> **1. Re: We could view this as a voice conversion setup, but with the additional task of rendering translated voice in a different language**
>
> The primary goal of S2ST is to translate the source speech faithfully and naturally into speech in the target language. Cross-lingual voice transferring is an optional add-on feature for S2ST. In this work, we made significant contributions to both of them.
>
> It is unclear to us how this is related to a voice conversion setup. We would appreciate clarifications from the reviewer.
>
> **2. Re: The novelty of concepts. It seems that most of the ideas are already there in Translatotron, and this paper modifies architecture (and does not give much insight into these changes).**
>
> We revised the beginning of Section 3 to add the motivation of the architecture design of Translatotron 2, especially the weaknesses in the original Translatotron that we addressed.
>
> We also added ablation studies in Appendix D. The results showed that simply replacing the corresponding components in the original Translatotron with the components from Translatotron 2 (i.e. Conformer, SpecAugment, and NAT decoder) does not explain the large performance improvement of Translatotron 2. Instead, such large performance improvement comes from the novel high-level architecture of Translatotron 2 (i.e. Figure 1 (a)).
>
> **3. Re: The presentation in section 4.1 - of retaining voice. “Wasn't the aim of the work to avoid using a speaker encoder for antispoofing purposes? Please clarify.”**
>
> In the proposed approach, unlike in the original Translatotron, the speaker encoder is only used when creating (i.e. synthesizing) the training dataset, but is not used in the Translatotron 2 model itself. As the result, for an original Translatotron model deployed in production, an attacker may exploit it for creating spoofing audio by feeding a speech with harmful content to the model as input, as well as feeding a reference speech in the voice to be mimic, so that the model would produce a speech with harmful content in the target speaker’s voice in the target language. Such attack is infeasible in Translatotron 2 because there is no such speaker encoder in the model. These are explained in Section 4. We also revised and added Appendix E to help clarify.

---

### Official Review · Reviewer_UaLy · 2021-11-08

**Correctness:** 4
**Technical Novelty And Significance:** 2
**Empirical Novelty And Significance:** 2
**Recommendation:** 6
**Confidence:** 4

**Main Review:**

The main strengths of the paper include the follows.
(1)	Compared to other direct S2ST works which mostly focus on translation quality, this work also aims to improve perceptual quality such as naturalness of the translated speech, which is also very important to the overall quality and user experience of a S2ST system.
(2)	Although Translatotron 2 also uses an encoder (conformer)-decoder (a stack of LSTMs) with attention modules, which is similar to a ST model, the difference is that instead of predicting subword units as a standard ST model, Translatotron 2 uses this framework to predict target phonemes, and then uses the phonemes as input to the spectrogram synthesizer. Note that the original Translatotron uses both source and target side phoneme predictions as auxiliary training tasks but Translatotron 2 only predicts and uses target side phonemes, yet achieving a better BLEU than Translatotron, especially on small dataset.
(3)	Experimental results showed Translatotron2 improves BLUE and naturalness (MOS) and robustness (UDR), compared to Translatotron, on conversational and smaller datasets, and also significant BLEU improvement (on ASR transcripts) on the multi-lingual CoVoST 2 dataset.

The main weaknesses of the paper include the follows.
(1)	The innovations from Translatotron 2 over Translatotron are not very strong. The major innovations are the use of the encoder(conformer)-decoder(LSTM) with attention framework to predict target phonemes, instead of using both source and target phonemes, and use phonemes as input to the spectrogram synthesizer.  The synthesizer in Translatotron 2 replaces the attention module with a duration-based upsampler, which yields more robust synthesized speech. But this approach is similar to the previous published works such as (Shen et al., 2020). And, the synthesizer in Translatotron 2 is simplified as it only used the L2 loss on the total predicted duration of the entire sequence.  The more principled approaches such as using per-phoneme duration labels for training the duration predictor and applying FVAE alignment, as in (Shen et al., 2020), would be important to be investigated in this work, but they were left for future work.

(2)	Some experimental results need to be further discussed. For example, the effect of ConcatAug in Table 2, Table 3, and Table 4 is not discussed clearly. For example, ConcatAug hurts BLEU, MOS and UDR for the relatively larger conversational dataset, and it is useful to provide analysis for it.

(3)	Although there is a significant improvement from Translatotron 2 over Translatotron on translation quality, naturalness and robustness of predicted speech, as can be seen from Table 2, there is still a relatively significant gap between Translatotron 2 and cascaded (ST->TTS). It would be useful to provide analysis and provide insights on more focused future directions on direct S2ST.


**Summary Of The Paper:**

This paper advances the previously proposed direct speech-to-speech translation (S2ST) model Translatotron.  Direct S2ST has many benefits but has quite limited research effort. The major contributions of this work can be summarized as follows. Translatotron 2 addressed the following major problems of Translatotron: 1)  translation quality, naturalness and robustness of predicted speech are significantly improved and now comparable to a cascaded system (i.e., speech-to-text translation followed by TTS).  2) Similar to Translatotron, the model retrains source speaker’s voice in predicted speech, but Translatotron 2 achieves this without relying on any explicit speaker embedding or speaker ID, hence Translatotron 2 will not be able to generate speech in a different speaker’s voice. This will address some ethical concerns for deployment. Translatotron 2 achieved these improvements through the following approaches. Compared to the previous Translatotron model,  1) Translatotron 2  uses  the output from the auxiliary target phoneme decoder as an input to the spectrogram synthesizer;  2) the spectrogram synthesizer is duration-based, while still keeping the benefits of the attention mechanism.

**Summary Of The Review:**

This paper extends the previous direct S2ST Translatotron model with Translatotron 2 model, which significantly improves translation quality (BLEU), the naturalness (MOS) and robustness (UDR) of predicted speech compared to Translatotron,  through a few modifications. The paper also proposed an approach to train targets in source speaker’s voice and prohibits generation in a different speaker’s voice to mitigate spoofing audio artifacts.  Experimental results on S2ST datasets of different sizes (relative large and small) and multilingual S2ST demonstrate the significantly improved performance from Translatotron 2 over Translatotron, and also the effectiveness of the proposed approach to retrain the source speaker’s voice in the translated speech.

---

> ### Author Response · Authors · 2021-11-16
> **Responses**
>
> We thank the reviewer for the helpful feedback. Below are the itemized responses regarding each comment. We have also incorporated these into the revised version.
>
> **1. Re: The innovations from Translatotron 2 over Translatotron**
>
>  - We added ablation studies in Appendix D. As the results show, while the use of Conformer, SpecAugment, and NAT decoder does help the performance of Translatotron 2, the contribution from these choices is minor. The performance of Translatotron 2 primarily comes from the novel model architecture (Figure 1 (a)), rather than the architectural choices of the individual components (i.e. encoder, decoder, and synthesizer). These results endorse the novelty of our model architecture contributions.
>  - This work is the first direct S2ST model that achieved comparable performance (both translation quality and speech quality) to cascade systems (on 3 datasets, including both bilingual and multilingual). Furthermore, it’s the first S2ST model that is able to preserve each speaker’s voice on speaker turns. We believe that such results are important breakthroughs and will encourage the community on the research on direct S2ST, which is a research topic rarely touched in the past. We believe that these results justify strong empirical contributions.
>
> **2. Re: Left using per-phoneme duration labels or applying FVAE alignment in synthesizer training as future works**
>
> While we believe such approaches would improve the performance of Translatotron 2, we consider such improvements to be incremental and likely minor, compared to the large improvement of Translatotron 2 over the original Translatotron. In the meantime, both such approaches bear significant cost: 1) using per-phoneme duration labels would require such labels available in the training data, which can be expensive to obtain; 2) adding FVAE would make the model architecture significantly more complicated.
>
> We conducted preliminary experiments of using FVAE in the synthesizer training. We haven’t seen clear translation quality win from it, which is not surprising because it has no direct impact on the speech-to-phoneme translation objective during training. We are expecting it to show benefits in the speech generation quality, but haven’t run formal evaluation due to the long turnaround and the cost of subjective evaluation. This preliminary result aligns with our judgment that the potential improvement to be incremental.
>
> **3. Re: The effect of ConcatAug in Table 2, Table 3, and Table 4 is not discussed clearly**
>
> We revised and added such discussion at the end of Sec. 5.4.1.
>
> **4. Re: Analysis and insights regarding the remaining quality gap between Translatotron 2 and cascade S2ST**
>
> We revised and added such discussion at the end of Sec. 5.1.

---

> > ### Comment · Reviewer_UaLy · 2021-11-30
> > **Responses to Author Rebuttal**
> >
> > Thank the authors for clarifying the questions in my review and updating the paper.  The rebuttal and updated draft addressed some of my questions and concerns.
> >
> > However, some important concerns still remain:
> >
> > 1. The innovation of the paper and the justification for the innovation is not convincing.
> > (1) The authors claimed the architectural choice as shown in Figure 1(a) is the major innovation of Translatotron 2 over Translatotron, as discussed in Appendix D and motivated in Section 3. But the ablation study to prove this point is only conducted by replacing the Conformer encoder with a RNN-based encoder, and replacing the auto-regressive synthesize with a Conformer-based non-autoregressive synthesizer.  Instead of analyzing alternative components in Translatotron 2, it would be more convincing to justify the contributions of the architectural choice, by conducting ablation study on alternative architectures to Figure 1(a), in order to prove the importance of designing the target phoneme decoder and a target mel-spectrogram synthesizer, and the importance of using a single attention module.
> >
> > (2) The ablation study in Table 9 introduced new questions:
> > (a)  I think SpecAugment is applied to both Translatotron 2 and Translatotron in Table 9, as it shows  "Translatotron w/ SpecAugment", but the results for Translatotron in Table 9 are all worse compared to their results in Table 5. This is confusing.
> > (b) Table 9 shows that SpecAugment, which was proposed in other earlier work and widely used, contributes a significant gain to Translatotron 2.
> >
> > 2. The effect of ConcatAug, introduced as one minor innovation of the paper, is mixed. The authors observed that while ConcatAug is effective to support voice retention on speaker turns, and can further improve the translation quality and the speech naturalness on models with lower performance (e.g., low-resource settings), it may negatively impact the speech naturalness and similarity on models with strong performance, as shown in Table 2 and Table 3, where ConcatAug hurts BLEU, naturalness and similarity. The authors did not provide insights to explain this or future directions to improve this approach.
> >
> > 3. Compared to cascaded system (ST->TTS), as shown in Table 2, there is still a large gap between Translatotron 2 and cascaded system performance. So it is not fair to claim that Translatotron 2 has achieved comparable performance to cascaded systems.
> >
> > 4. I still think It is important to explore the two directions of per-phoneme duration labels or applying FVAE alignment in synthesizer training, in this work instead of being left to future work, and compare to their effects on translation quality and speech generation quality.
> >
> > Based on these concerns, I will keep my current recommendation score.

---

> > > ### Author Response · Authors · 2021-11-30
> > > **Reply**
> > >
> > > We thank the reviewer for the reply. We believe that there are misunderstandings which we would like to clarify. We would appreciate if the reviewer can take the following further clarification into consideration.
> > >
> > > **Re: 1.(1): It would be more convincing to justify the contributions of the architectural choice, by conducting ablation study on alternative architectures to Figure 1(a).**
> > >
> > > It is not clear what kind of alternative architectures that the reviewer wanted to see. It would be helpful if the reviewer can clarify. Like almost every neural network models, there are infinite possibilities of "alternative architectures", some of which are established, some of which are yet to be discovered, and most of which are simply ineffective. We already included comparison to all known established alternative architectures in Table 2, 3, 4, and 5.
> > >
> > > **Re: 1.(2)(a): The results for Translatotron in Table 9 are all worse compared to their results in Table 5.**
> > >
> > > Table 9 Row 2 is identical to Table 5 Row 2, which correspond to matching model architecture. Table 9 Row 1 and Row 3 are worse than Table 5 Row 2, and that is exactly the value of the ablation.
> > >
> > > **Re: 1.(2)(b): Table 9 shows that SpecAugment ... contributes a significant gain to Translatotron 2.**
> > >
> > > As Table 9 shows, SpecAugment contributes 1.1 BLEU on average on the 4 language pairs to Translatotron 2. As a comparison, the performance difference between Translatotron 2 and the original Translatotron is 8.4 BLEU on average, when SpecAugment is already applied to the original Translatotron. This result endorses that the performance improvement primarily comes from the novel high-level architecture, rather than using SpecAugment in Translatotron 2.
> > >
> > > **Re: 2: The effect of ConcatAug ... is mixed (on BLEU evaluation).**
> > >
> > > While its impact on BLEU is mixed, it enables the voice retention on speaker turns. At the best of our knowledge, this is the first ever approach achieved so.
> > >
> > > **Re: 3: As shown in Table 2, there is still a large gap between Translatotron 2 and cascaded system performance. So it is not fair to claim that Translatotron 2 has achieved comparable performance to cascaded systems.**
> > >
> > > As shown in Table 2, we reduced the BLEU gap between direct S2ST and cascade S2ST from 16.4 to 3.0 on the Fisher dataset, and from 8.4 to 3.2 on the Conversational dataset. Therefore, we believe it is fair to say that this work brings the performance of direct S2ST to be comparable to cascade S2ST. Note that we did not claim that their performance are on-par.
> > >
> > > **Re: 4: It is important to explore the two directions of per-phoneme duration labels or applying FVAE alignment in synthesizer training, in this work instead of being left to future work.**
> > >
> > > We reserve our view that these are optional add-ons to our work, but not essential to it. Our preliminarily experimental results are consistent with this view.

---

### Author Response · Authors · 2021-11-16
**New revision uploaded**

We thank all reviewers for the thoughtful comments and constructive suggestions, which were extremely helpful to us in improving the paper. We have uploaded a revised version to address these comments.

The major changes include:
 - An ablation study is added in Appendix D.
 - A description of the cross-lingual voice transferring TTS model used in Sec 4.1 is added in Appendix E.
 - A discussion on the motivation of the architectural design of Translatotron 2 as well as the weaknesses of the original Translatotron being addressed is added at the beginning of Section 3.
 - More discussion on the ConcatAug experiment results is added in Sec 5.4.1
 - More analysis of the translation quality gap and potential directions for improvement is added in Section 5.1
 - Many minor revisions to clarify the comments from the reviewers.
 - Layout changes in order to stay within the page limit after the above revisions.

---

> ### Author Response · Authors · 2021-11-19
> **Another revision uploaded**
>
> We have uploaded another revision to update the numbers in Table 9 row 1 in Appendix D after this baseline is fully converged. As noted in footnote 5 in the previous revision, the numbers in this row got a small increase, which does not affect the conclusions as anticipated.

---

### Decision · Program_Chairs · 2022-01-20

**Decision:**

Reject

**Comment:**

The paper proposed a speech-to-speech translation (S2ST) model. The model is trained end-to-end from speech to speech, along with an auxiliary speech-to-phoneme task. Experiments firmly support multiple claimed improvements to the previous model.

However, most reviewers argue the novelty and clarity of this paper, making this paper cannot be accepted by ICLR-2022. We hope the authors can modify this paper accordingly.